# Effectiveness of Respiratory Muscle Training for Pulmonary Function and Walking Ability in Patients with Stroke: A Systematic Review with Meta-Analysis

**DOI:** 10.3390/ijerph17155356

**Published:** 2020-07-24

**Authors:** Diana P. Pozuelo-Carrascosa, Juan Manuel Carmona-Torres, José Alberto Laredo-Aguilera, Pedro Ángel Latorre-Román, Juan Antonio Párraga-Montilla, Ana Isabel Cobo-Cuenca

**Affiliations:** 1Faculty of Physiotherapy and Nursing of Toledo, University of Castilla-La Mancha, 45005 Toledo, Spain; dianap.pozuelo@uclm.es (D.P.P.-C.); juanmanuel.carmona@uclm.es (J.M.C.-T.); anaisabel.cobo@uclm.es (A.I.C.-C.); 2Multidisciplinary Research Group in Care (IMCU), University of Castilla-La Mancha, 45005 Toledo, Spain; 3Social and Health Care Research Center (CESS), University of Castilla-La Mancha, 16071 Cuenca, Spain; 4Maimónides Biomedical Research Institute of Córdoba (IMIBIC), 14004 Córdoba, Spain; 5Faculty of Health Sciences, University of Castilla-La Mancha, Talavera de la Reina, 45600 Toledo, Spain; 6Department of Didactics of Music, Plastic and Corporal Expression, University of Jaén, 23071 Jaén, Spain; platorre@ujaen.es (P.Á.L.-R.); jparraga@ujaen.es (J.A.P.-M.)

**Keywords:** stroke, respiratory muscle training, meta-analysis, walking ability, pulmonary function

## Abstract

*Background*: Neurological dysfunction due to stroke affects not only the extremities and trunk muscles but also the respiratory muscles. *Aim*: to synthesise the evidence available about the effectiveness of respiratory muscle training (RMT) to improve respiratory function parameters and functional capacity in poststroke patients. *Methods*: a systematic electronic search was performed in the MEDLINE, EMBASE, SPORTDiscus, PEDro and Web of Science databases, from inception to May 2020. *Study selection and data extraction*: randomised controlled trials (RCTs) that examined the effects of RMT versus non-RMT or sham RMT in poststroke patients. We extracted data about respiratory function, respiratory muscle strength and functional capacity (walking ability, dyspnea, balance, activities of daily life), characteristics of studies and features of RMT interventions (a type of RMT exercise, frequency, intensity and duration). Two reviewers performed study selection and data extraction independently. *Results*: nineteen RCTs met the study criteria. RMT improved the first second forced expiratory volume (FEV1), forced vital capacity (FVC), peak expiratory flow (PEF), maximal expiratory pressure (MEP), maximal inspiratory pressure (MIP) and walking ability (6 min walking test), but not Barthel index, Berg balance scale, and dyspnea. *Conclusions*: RMT interventions are effective to improve respiratory function and walking ability in poststroke patients.

## 1. Introduction

Stroke is a loss of focal neurological function due to infarction or haemorrhage in an essential part of the brain [1]. This neurological damage affects not only peripheral but also respiratory muscles, causing respiratory weakness, changes in the respiratory pattern and decreases in respiratory volumes and flows [2,3]. These breathing alterations are related to a decrease in physical activity and therefore, to a reduction in the ability to carry out the activities of daily life [4].

Respiratory muscles respond to training similarly to any other skeletal muscle, so just as the upper and lower limb muscles are trained in stroke patients, the respiratory muscles must be trained. In this regard, respiratory muscle training (RMT) consists of repetitive breathing exercises with hand-held respiratory trainer devices to provide pressure threshold or flow-dependent resistance against inhalation (inspiratory muscle training (IMT)) and/or exhalation (expiratory muscle training (EMT)) [5,6] to stimulate this musculature and to produce changes in the muscles’ structure.

Previous systematic reviews and meta-analyses have synthesised the available evidence about the effectiveness of RMT exercises on improving respiratory function [3,7,8,9,10], even exercise tolerance in poststroke patients; however, some weaknesses of these reviews were that they did not only include randomised controlled trials (RCTs) [10], and that they included a limited number of studies [3,8]. Moreover, since the last systematic review and meta-analysis [7], many studies aiming at analysing the effect of RMT on respiratory function and functional capacity have been published.

For these reasons, this systematic review and meta-analysis aimed to synthetize the most novel evidence about the effectiveness of RMT to improve respiratory function, respiratory muscle strength and functional capacity in poststroke patients.

## 2. Materials and Methods

This meta-analysis was performed in accordance with the Preferred Reporting Items for Systematic Reviews and Meta-Analyses (PRISMA) [11] and following the recommendations of the Cochrane Handbook for Systematic Reviews of Interventions [12]. Additionally, this systematic review and meta-analysis was registered through PROSPERO (awaiting register number; ID: 182082).

### 2.1. Search Strategy

Two researchers (DPP-C and AIC-C) independently searched MEDLINE (via Pubmed), EMBASE (via Scopus), SPORTDiscus, PEDro and Web of Science databases from inception to May 2020; disagreements were solved by consensus or involving a third researcher (JAL-A). The search strategy used was as follow: (“inspiratory muscle training” OR “IMT” OR “expiratory muscle training” OR “respiratory therapy” OR “chest physiotherapy” OR “respiratory exercise” OR “breathing exercises” AND (“stroke” OR “acute stroke” OR “cerebral stroke”) AND (“pulmonary function” OR “respiratory function” OR “exercise tolerance” OR “walking ability” OR “gait ability”) AND (random* control* trials). The literature search was complemented by scanning the reference list of the included articles, and the list of references of previous systematic reviews was reviewed for additional relevant studies.

### 2.2. Study Selection

Reviewers were not blinded to authors, journals, or institutions. Included articles were RCTs studies that analysed the effectiveness of RMT (IMT and/or EMT) on improving pulmonary function, respiratory muscle strength and functional capacity parameters in patients with stroke.

The inclusion criteria were: (1) Patients: adults with stroke (haemorrhagic or ischaemic); (2) Intervention: IMT, EMT or both; (3) Control: no respiratory training or sham respiratory muscle training (without any resistance); (4) Outcomes: variables of pulmonary function such as the first second forced expiratory volume (FEV1), forced vital capacity (FVC), and peak expiratory flow (PEF); parameters of respiratory muscle weakness: maximal expiratory pressure (MEP) and maximal inspiratory pressure (MIP); Functional capacity including 6 min walking test (6-MWT), Barthel index (MBI), Berg balance scale (BBS), and dyspnea (Borg scale); (5) type of studies: RCTs.

Exclusion criteria were: (1) studies not written in English or Spanish; (2) studies not reporting the outcome variables; (3) non-RCTs; (4) insufficient data.

### 2.3. Data Extraction

For data extraction, two authors (DPP-C and AIC-C), independently, used a standardised data collection form. From each selected study, the following data were collected: (1) first author’s name and year of publication; (2) country; (3) period of data collection; (4) characteristics of study sample: sample size, age mean, gender; (5) data concerning inclusion/exclusion criteria; (6) main study outcomes. We emailed the corresponding author requesting the data when some information was lacking.

### 2.4. Risk of Bias Assessment

While in the Prospero register we stated that the intent to use the Cochrane Collaboration tool to assess the risk of bias (RoB2) [13], we observed that all previous systematic reviews and meta-analysis on this topic had used the PEDro Scale to assess the quality of included studies. Thus, with the aim of increase the homogeneity and the comparability, we decided to use this scale instead of RoB2.

PEDro Scale is a useful tool for assessing the quality of physical therapy trials [14]. This scale has 11 items designed for rating the methodological quality of RCTs assessing the internal validity (e.g., random and concealed allocation, baseline similarity, blinding, and intention-to-treat analysis) and statistical reporting. The total PEDro score ranges from 0 to 10 points (the first item, eligibility criteria, is not included in the total score); higher score means better study’s quality [15].

Data extraction and risk of bias assessment were independently completed by two reviewers (DPP-C and AIC-C) if appeared inconsistencies were solved by consensus or involving a third researcher (JAL-A).

### 2.5. Statistical Analysis

For each of the main outcomes (FEV1, MIP, MEP, PEF, FVC, 6-MWT, dyspnea, MBI and BBS), a separate pooled estimate of effect size (ES) and their respective 95% confidence intervals (95% CI) was calculated. The ES parameters from preintervention to postintervention between groups (exercise intervention vs control) [16] in each study were calculated assuming a correlation coefficient of 0.5 and using a random-effects model based on the DerSimonian and Laird method [17]. Statistical heterogeneity was analysed by the I2 statistic. Heterogeneity was considered as not important (I2: 0% to 40%), moderate (I2: 30% to 60%), substantial (I2: 50% to 90%) or considerable (I2: 75% to 100%); the corresponding *p*-values were considered [12]. When a study reported results separately for IMT and EMT, we calculated a combined estimate.

According to the Cochrane Handbook recommendations, when mean and standard deviation (SD) were lacking, and available statistics were median and interquartile range (IQR), the IQR was divided by 1.35 to transform these estimates on CI [18]. Finally, when studies were scaled inversely (i.e., lower values indicated worse outcomes), the mean in each group was multiplied by −1.

Subgroup analyses were conducted when it was possible due to the number of studies, to assess whether the intervention of RMT (IMT/EMT) or only IMT allowed better results on outcome variables. Sensitivity analyses were performed by removing studies one by one to assess the robustness of the summary estimates and to detect whether any study accounted for a large proportion of heterogeneity among RMT and ES pooled estimations. Meaningful results of the sensitivity analyses were considered when the resulting estimates were modified beyond the CIs of the original summary estimate.

Finally, Egger regression asymmetry method and visual inspection of funnel plots were used to assess publication bias. Statistical analyses were performed using Stata Statistical software, version 16.0 (Stata, College Station, TX, USA).

## 3. Results

The electronic search retrieved 826 studies, and we identified two new studies through manual search. After removing duplicated studies, the title and abstract of 800 studies were revised. Of these, 754 were excluded by irrelevancy and 46 were full-text revised. Finally, nineteen studies aimed to assess the effectiveness of RMT on pulmonary function and functional capacity variables in patients with stroke were included in the meta-analysis [19,20,21,22,23,24,25,26,27,28,29,30,31,32,33,34,35,36,37]. Figure 1 shows the flow of study selection. The reasons for the full-text studies excluded are summarised in Appendix A.

Table 1 depicts the characteristics of the included studies and the participants. Studies were conducted in Korea [20,22,23,24,25,26,27,33,34,36], Turkey, Spain [28,30], Brazil [21,35], Germany [32], United Kingdom [31] and Taiwan [19,29].

The nineteen studies included involved 643 patients with stroke, with the sample size varying between 12 and 82 participants. The mean age of participants was 61.3 years.

All included studies performed RMT; nevertheless, ten studies carried out IMT and EMT [19,20,21,23,28,30,31,32,33,36], and nine studies only performed IMT [22,24,25,26,27,29,34,35,37]. Regarding the threshold devices used in the RMT interventions, they were very varied: Respironics, Respifit-S, Orygen Dual Valve, SpiroTiger, Threshold, PowerBreath, tri-ball incentive spirometer and Dofin Breathing (Table 1).

The interventions were very diverse; the number of sets per session varied between two and ten sets; the repetitions in each set ranged between five and 30. Other studies listed time rather than the number of repetitions; the load of RMT in the majority of studies was at 30% of MIP/MEP at the beginning, increasing with the intervention. However, some studies used higher loads at 40% or 50% at the beginning of the intervention.

The length of the interventions ranged between three and ten weeks. In the control group, most studies conducted the conventional stroke rehabilitation program, however, four out of twenty studies performed sham respiratory training without resistance and/or progression [21,30,31,35].

### 3.1. Risk of Bias Assessment

The risk of bias of included studies was assessed using the PEDro Scale. All RCTs included complied with the following items: random allocation, to report between-group differences and in to report point estimate and variability (Table 2). The highest score was eight points for two studies [21,30]. Four studies scored seven [19,31,35,37], mainly due to the non-blinding of therapist and assessors and the non-presentation of an intention-to-treat analysis. Five studies scored six and four studies obtained five points. The lowest score was three points for the study of the NJ Jung et al. [25]. The three resting studies obtained four points [24,33,34]. Thus, fifteen studies obtained higher or equal to five points, a moderate quality.

### 3.2. Effect of Intervention: Pooled Estimates

#### 3.2.1. Pulmonary Function

Among nineteen studies included, eleven displayed results about FEV1 [19,20,23,25,27,29,32,33,34,36,37]. When all RMT interventions (IMT/EMT and IMT) were jointly analysed, the summary ES was 0.57 (95% CI: 0.24 to 0.90; I^2^ = 42.9%, *p* = 0.064) (Figure 2). The subgroup analysis by type of intervention showed a slightly greater effect in the pooled of the studies that performed only IMT [ES= 0.89 (95% CI: 0.19 to 1.58; I^2^ = 66.3%, *p* = 0.018)] compared with the IMT/EMT subgroup [ES= 0.38 (95% CI: 0.06 to 0.69; I^2^ = 0.0%, *p* = 0.650)].

The FVC was analysed in nine studies [19,23,25,27,29,32,34,36,37]. The pooled ES was 0.32 (95% CI: 0.06 to 0.58; I^2^ = 0.0%, *p* = 0.507) (Figure 3). The subgroup analysis by type of RMT exercises did not show great differences between the ES of both groups, although without statistical significance in any group (IMT and IMT/EMT).

Other pulmonary function parameters such as PEF and the strength of inspiratory and expiratory muscles (MIP and MEP) were analysed in the studies included. Seven studies showed results about PEF and the pooled estimated ES was 0.48 (95% CI: 0.20 to 0.77; I^2^ = 0.0%, *p* = 0.823) (Figure 4) [20,23,27,33,34,36,37], nevertheless, the subgroup analysis displayed a statistically significant ES in the IMT/EMT group (ES = 0.55; 95% CI: 0.17 to 0.92) but not in the IMT group (ES = 0.40; 95% CI: −0.04 to 0.84).

Equally, MEP was analysed in seven studies [19,20,21,28,30,31,37], and only one of these performed exclusively IMT exercises [37]; thus, the overall ES was 0.55 (95% CI: 0.12 to 0.98; I^2^ = 71.9%, *p* = 0.002) (Figure 5).

The MIP was examined in nine studies [19,20,21,22,28,30,31,35,37], and obtained a pooled ES= 0.84 (95% CI: 0.44 to 1.24; I^2^ = 69.2%, *p* = 0.001) (Figure 6); three of these studies performed only IMT exercises [22,35,37], and six studies carried out both IMT and EMT exercises, showing statistically significant improvements for both groups of interventions.

Sensitivity analyses for all pulmonary function variables showed that the pooled ES estimates were not significantly modified in magnitude or direction when individual study data were removed from the analysis one at a time. The exception was the FVC analysis, which lost the statistical significance when the studies of KM Jung et al. [25] and K Kim et al. [36] were removed from the analysis.

There was no evidence of publication bias by funnel plot asymmetry and Egger’s test in all pulmonary function variables studied.

#### 3.2.2. Functional Capacity

Six out of nineteen studies evaluated the 6-MWT [21,22,24,25,26,33]. Of these, two studies only performed IMT exercises [25,26]; for this, a subgroup analysis by type of RMT exercises was not performed. The overall ES for the 6-MWT was 0.39 (95% CI: 0.05 to 0.74; I^2^ = 0.0%, *p* = 0.919) (Figure 7). When the studies were removed one by one, the pooled ES analysis showed a loss of statistical significance when the study of NJ Jung et al. [26] was removed.

Dyspnea was evaluated in five studies [19,21,29,33,37]; due to the limited number of studies, the subgroup analysis by type of RMT exercise was not performed. The pooled ES of RMT interventions on dyspnea was 0.47 (95% CI: −0.30 to 1.23; I^2^ = 77%, *p* = 0.002) (Figure 8), showing a positive trend of improvement which was not statistically significant.

The MBI [19,23,29] and BBS [23,24,27] were only evaluated in three studies. In both, the RMT interventions did not show a statistically significant improvement (MBI: 0.22 (95% CI: −0.31 to 0.75; I^2^ = 37.6%, *p* = 0.201; and BBS: 0.06 (95% CI: −0.37 to 0.49; I^2^ = 0.0%, *p* = 0.953).

Publication bias was not assessed by funnel plot and Egger test, due to the reduced number of studies that evaluated these variables of functional capacity [38].

## 4. Discussion

In the present study, the results show a significant positive relationship between the RMT interventions on the improvement of pulmonary function parameters (FEV1, FVC, PEF, MEP, MIP) and the functional capacity such as walking ability (6-MWT) but not in balance (BBS), MBI and dyspnea, in patients with stroke.

The subgroup analyses by type of RMT exercises did not show differences between the exclusive IMT interventions and the IMT/EMT interventions, except in the PEF, in which the combined IMT and EMT exercises showed a greater effect than only IMT exercises on the improvements of this pulmonary parameter.

A previously published meta-analysis that included eleven trials also obtained similar results [7]. Nevertheless, the main difference was found in dyspnea. Contrary to this study, our meta-analysis did not find a positive effect of RMT on dyspnea in patients with stroke. These differences could be due to several reasons: first, the previous study only included two moderate methodological quality RCTs, a reduced number of studies to provide consistent conclusions about the effectiveness of RMT on dyspnea; second, patients with stroke usually have a low perception of dyspnea, due to their dissociation between respiratory effort and dyspnea [39].

Regarding the pulmonary function parameters, our results showed positive effectiveness of RMT interventions to improve FEV1, FVC, PEF, MEP, and MIP in patients with stroke. These results are in line with previous systematic reviews and meta-analyses results [7,8]. Patients who have suffered a stroke present abdominal and diaphragm dysfunction, causing a decrease of respiratory muscle strength [2,40,41]. This weakness of respiratory muscles is usually associated with reduced lung volumes, flows and restrictive ventilatory patterns [42]. Thus, it seems obvious, and the results of this meta-analysis corroborate this, that the training of the respiratory muscle will improve the lung volumes and flows, noting an increase of FEV1, FVC, PEF, MEP and MIP.

Aiming to facilitate the clinical interpretation of these results, we expressed the pooled effect estimate of each pulmonary function parameter on clinical measurement improvements, using methods recommended by the Cochrane Collaboration [43]. Thus, our data show that RMT interventions increase baseline FEV1 values in 12.2% of predicted-FEV1 by age and sex, FVC baseline values improve in 6.75% of predicted FVC, PEF baseline measure increases in 46.97 litres per second. Finally, MEP and MIP improved their baseline values in 10.05 and 22.40 cm H_2_0, respectively. These results align with previously published meta-analysis [7,8,9,10].

The improvement of inspiratory and expiratory muscle strength had significant implications for the functional capacity of poststroke patients. The weakness of respiratory muscles, secondary to stroke, can lead to a reduced tolerance to exercise [8,44]. Thus, the benefits gained by the increase of respiratory muscle strength and lung function could improve the tolerance to exercise [8].

Our results showed that RMT achieved an improvement of walking ability measured by 6-MWT, with an ES= 0.39, which when translated in terms of clinical measure improvement, using methods endorsed by the Cochrane Collaboration [43], is equal to an increase of 25 m travelled in 6 min. Although it could be seen as little progress, walking ability is a significant predictor of physical activity and social and community participation after stroke [45,46], so even small improvements could imply immense benefits for the physical and social health of stroke patients. They can help these patients to carry out daily activities more efficiently.

In line with this, three studies reported results about the Barthel index, a scale used to measure disability or dependence in activities of daily living [47]. The pooled estimate showed a positive but not statistically significant trend. The same occurred with the BBS [48]. Nevertheless, the limited number of studies that reported these variables do not allow for robust conclusions about it.

Despite the wide variety of characteristics of RMT interventions, the previous meta-analysis concluded that 30 min of RMT, five times/week, for five weeks could be sufficient to increase respiratory muscle strength in very weak poststroke patients [3].

This study has several limitations that must be considered, some of them, such as the heterogeneity, are inherent to meta-analyses. First, although a higher number of studies than the previous meta-analysis was included, not all of them report the same variables, so some variables such as MBI or BBS only were informed in three studies. Second, it is known that patients with more significant respiratory muscle weakness usually respond better to RMT [8], and in the present work around the half studies [19,20,21,22,28,29,30,31,37] included poststroke patients with respiratory muscle weakness (MIP < 50 cm H_2_O) which could have influenced the results. Third, the inclusion/exclusion criteria were different in the studies included, equally to the RMT characteristics of interventions. Nevertheless, the subgroup analysis by type of RMT exercises allowed for the comparison of the effect of IMT alone with the IMT and EMT exercises. Fourth, despite the number of studies included, the total sample of participants included was scarce.

## 5. Conclusions

The RMT interventions are effective in improving pulmonary function parameters (FEV1, FVC, PEF), strength of expiratory and inspiratory muscles (MEP and MIP), and walking ability in poststroke patients. More well-designed RCTs with larger sample sizes are needed to examine the most appropriate features of interventions: IMT, EMT or both, duration, frequency, and intensity to establish the highest clinical efficacy.

## Figures and Tables

**Figure 1 ijerph-17-05356-f001:**
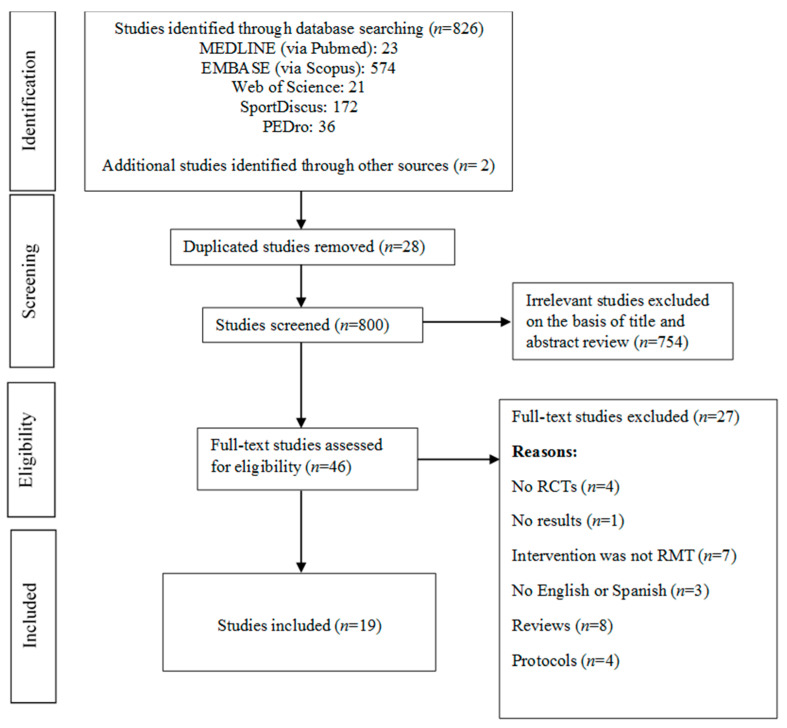
Literature search: Preferred Reporting Items for Systematic Reviews and Meta-Analyses (PRISMA) consort diagram. RMT, Respiratory muscle training; RCTs, randomized controlled trials.

**Figure 2 ijerph-17-05356-f002:**
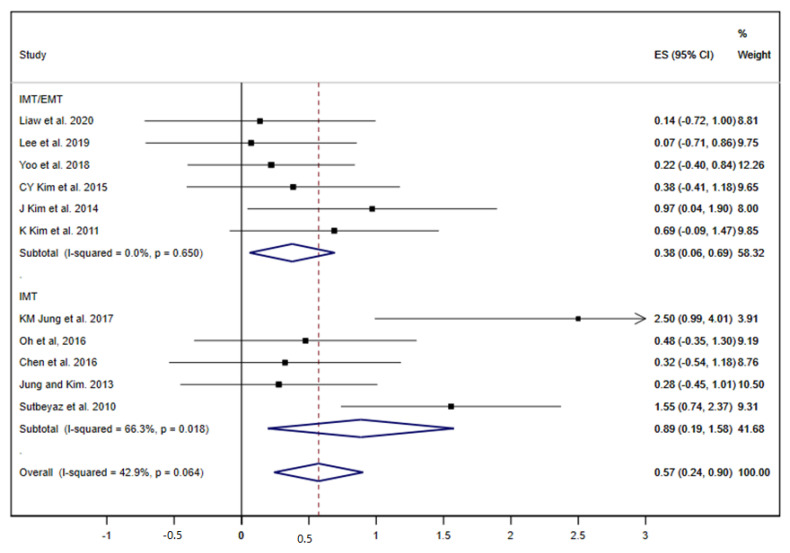
Forest plot showing the effect size (ES) of respiratory muscle training (RMT) on first second forced expiratory volume (FEV1) between intervention and control groups for each study. IMT, inspiratory muscle training; EMT, expiratory muscle training.

**Figure 3 ijerph-17-05356-f003:**
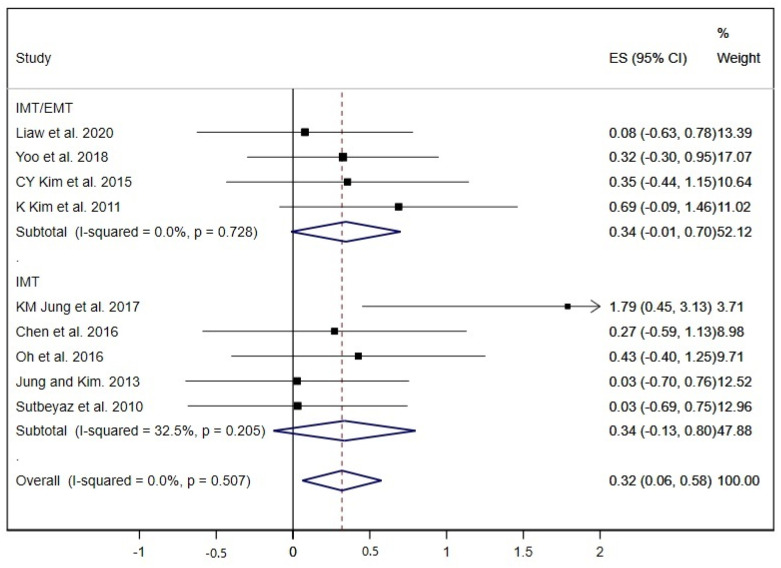
Forest plot showing the effect size (ES) of respiratory muscle training (RMT) on forced vital capacity (FVC) between intervention and control groups for each study. IMT, inspiratory muscle training; EMT, expiratory muscle training.

**Figure 4 ijerph-17-05356-f004:**
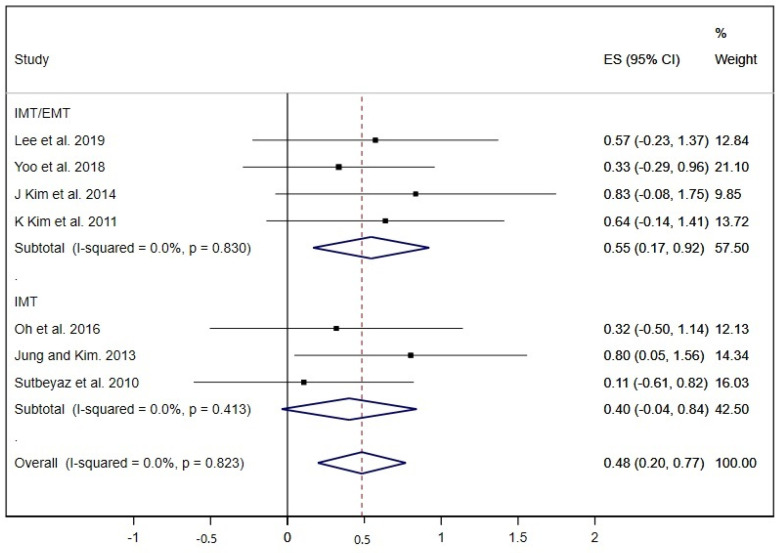
Forest plot showing the effect size (ES) of respiratory muscle training (RMT) on peak expiratory flow (PEF) between intervention and control groups for each study. IMT, inspiratory muscle training; EMT, expiratory muscle training.

**Figure 5 ijerph-17-05356-f005:**
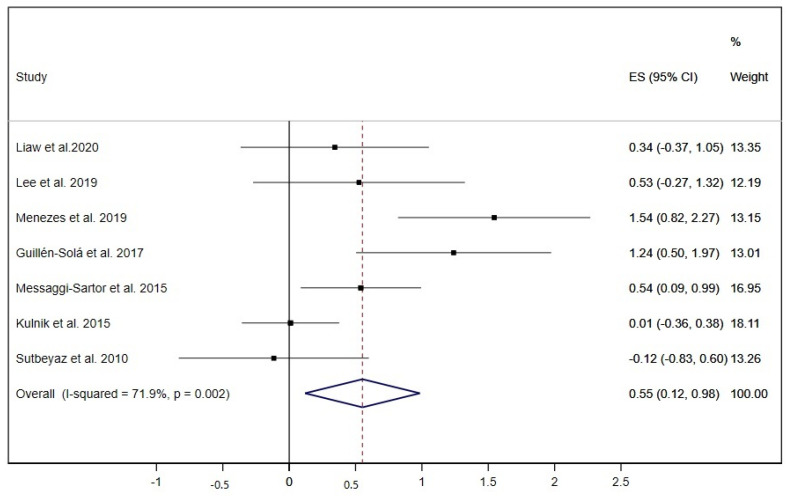
Forest plot showing the effect size (ES) of respiratory muscle training (RMT) on maximal expiratory pressure (MEP) between intervention and control groups for each study. IMT, inspiratory muscle training; EMT, expiratory muscle training.

**Figure 6 ijerph-17-05356-f006:**
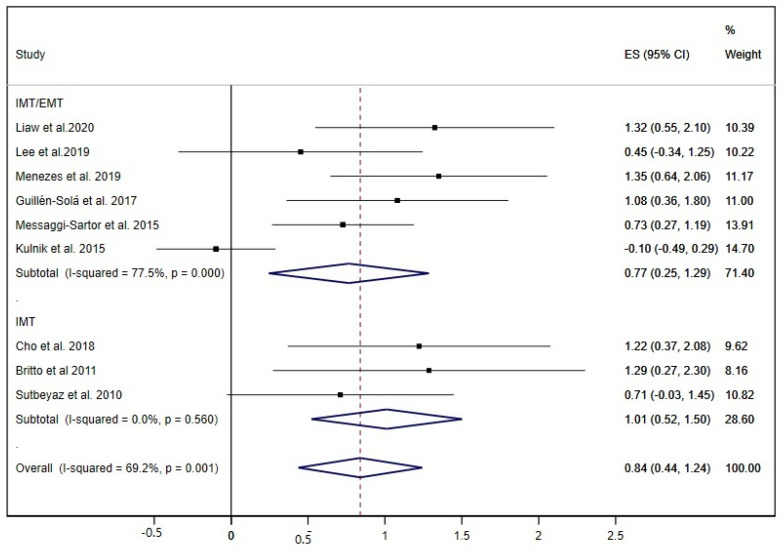
Forest plot showing the effect size (ES) of respiratory muscle training (RMT) on maximal inspiratory pressure (MIP) between intervention and control groups for each study. IMT, inspiratory muscle training; EMT, expiratory muscle training.

**Figure 7 ijerph-17-05356-f007:**
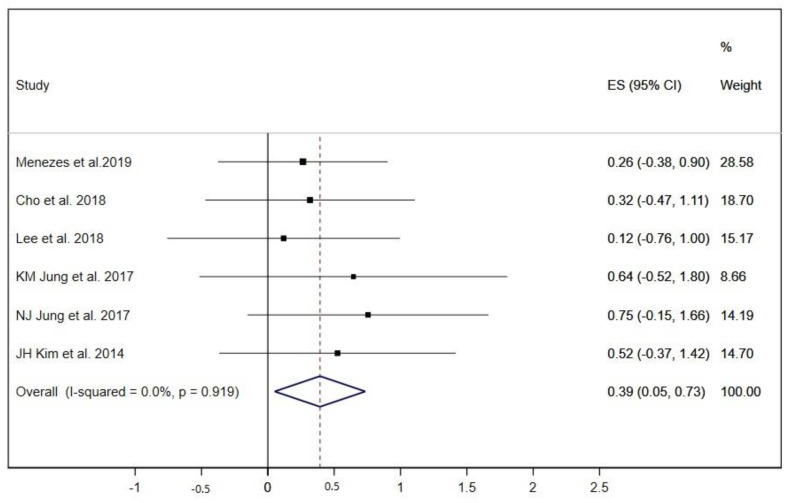
Forest plot showing the effect size (ES) of respiratory muscle training (RMT) on 6 min walking test (6-MWT) between intervention and control groups for each study. IMT, inspiratory muscle training; EMT, expiratory muscle training.

**Figure 8 ijerph-17-05356-f008:**
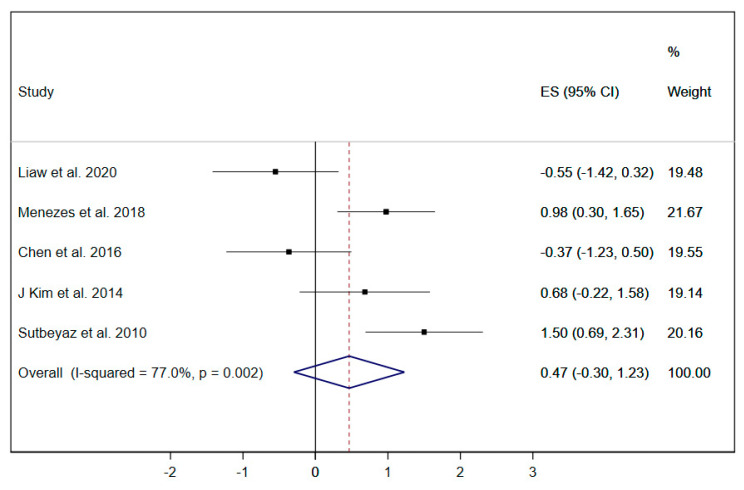
Forest plot showing the effect size (ES) of respiratory muscle training (RMT) on dyspnea between intervention and control groups for each study. IMT, inspiratory muscle training; EMT, expiratory muscle training.

**Table 1 ijerph-17-05356-t001:** Characteristics of studies included.

Study	Country	Participants/Mean Age	Inclusion/Exclusion Criteria	Intervention	Control	Study Outcomes
Liaw et al., 2020 [19]	Taiwan	*n* = 31 GI: 15 GC: 16 38.7% men 62.8 years	**Inclusion:** Aged 35 to 80 years with unilateral stroke > 6 months, respiratory muscle weakness, dysphagia, or dysarthria. **Exclusion:** Increased intracranial pressure, uncontrolled hypertension, decompensated heart failure, unstable angina, recent myocardial infarction, complicated arrhythmias, pneumothorax, bullae/blebs in the preceding 3 months, severe cognitive function or infection, recurrent stroke, brain stem stroke, and aphasia.	IMT load was 30%–60% of MIP (6 sets of 5 repetitions). EMT load was 15%–75% of MEP (5 sets of 5 breaths). *Session*: IMT and EMT, 1 to 2 times per day, 5 days a week for 6 weeks, 1 to 2 min between each set. *Device*: Dofin Breathing Trainer (DT 11 or DT 14). RHB program as CG.	RHB program: postural training, breathing control, improving cough technique, checking chest wall mobility, fatigue management, orofacial exercises, thermal tactile stimulation, Mendelsohn maneuvering, effort swallowing, or supra-glottic maneuver.	-FEV1 -MEP -MIP -MBI -Dyspnea -FVC
Lee et al., 2019 [20]	Seoul, Korea	*n* = 25 GI: 13 GC: 12 48% men 59.1 years	**Inclusion:** Stroke over 6 months, MMSE > 24, no facial palsy and receptive aphasia, and no prior thoracic or abdominal surgery. **Exclusion:** Medications for neuromuscular control or that provoke drowsiness, significant disability prior to stroke as evidenced by a score >3 on MRS, restrictive lung disease, TIS <10, and musculoskeletal problems in the pelvis or spine.	IMT/EMT at 30% of the resistance intensity on the first day of the week. *Session*: 10–15 times, 5 set for 20 min in a session and resting time of 30–60 s between each set. Trunk stabilization exercises: 40 min, 3 times a week for 6 weeks. *Device*: Threshold PEP; Threshold IMT-Respironics. RHB program as CG.	Conventional physical and occupational therapy conducted for 30 min, 2 times a day, and 6 times per week. Trunk stabilization exercises: 40 min, 3 times a week for 6 weeks.	-FEV1 -MEP -MIP -PEF
Menezes et al., 2019 [21]	Brazil	*n* = 38 GI: 19 GC: 19 50% men 63.5 years	**Inclusion:** Adults > 20 years, stroke > 3 months and < 5 years, MIP < 80 cm H_2_O or MEP < 90 cm H_2_O (respiratory muscle weakness), and not undertook any respiratory training. **Exclusion:** Patients with cognitive deficits, facial palsy, and/or any conditions.	Home based IMT/EMT. *Session*: 40 min/day (two 20 min sessions), 7 times/ week, over 8 weeks. Each 20-min session: 4 min sets of training, followed by 1 min rest. Load: 50% MIP/MEP *Device*: Orygen-dual valve.	Sham respiratory training without any resistance and progression.	-MIP -MEP -Dyspnea -6-MWT
Cho et al., 2018 [22]	Korea	*n* = 25 GI: 12 GC: 13 52% men 49.5 years	**Inclusion:** Adults > 20 years, stroke > 3 months, MIP < 70% of those predicted when adjusted for age and sex, had and were able to give informed consent and follow study procedures. **Exclusion:** Patients with facial palsy, myocardial infarction or acute heart failure within 3 months, pulmonary disease, neurological conditions or medications that interfere with neuromuscular control.	Hospital IMT. *Session*: 3 sets of 30 breaths, with a 1-min rest in between the sets. 5 days a week, for 6 weeks. Load: 30% MIP readjusted weekly. *Device*: PowerBreath K5. RHB program as CG.	RHB program: muscle strengthening exercises, Bobath therapy, gait training, and stair climbing training for 60 min/day, 5 days per week, for 6 weeks.	-MIP -Fatigue -6-MWT
Yoo et al., 2018 [23]	Seoul, Korea	*n* = 40 GI: 20 GC: 20 65% men 61 years	**Inclusion:** Adults >18 years; first episode of stroke within 3 months; moderate to severe stroke impairment; understand instructions and follow the study program. **Exclusion:** Persistent cardiopulmonary disease, coexisting brain disorders, uncontrolled hypertension, severe facial palsy or other oropharyngeal structural abnormality, severe oral apraxia, and tracheostomy.	Bedside IMT/EMT. *Session*: 10 min of breath stacking exercise + 10 min IMT + 10 min EMT; with a 5-min interval before each exercise. Twice/day for 7 days/week for a 3-wk period. *Device*: A flow-oriented incentive spirometer (Hyupsung) for IMT; Acapella vibratory PEP for EMT. RHB program as CG.	Conventional stroke RHB program: motion exercises, muscle strengthening, gait training, fine motor exercises, and activity of daily living training. It was performed for 30 min, twice a day 5 days a week, for 3 weeks.	-FVC -FEV1 -PEF -MBI -BBS
Lee et al., 2018 [24]	Jeonju, Korea	*n* = 20 GI: 10 GC: 10 50% men 60 years	**Inclusion:** Patients with hemiplegia due to stroke, 50–70 years, diagnosed 1–2 year previously, could perform 10-m walking independently and walk within 5–60 sec, ability to understand and follow the indication of the researcher. **Exclusion:** NA	Load of IMT: 50% of MIP increasing repetitions each week. *Session*: 20 min of IMT and 20 min of bracing exercise for 6 weeks and 5 days per week. Bracing exercise holding the pressometer at 40 mmHg. *Device*: POWERbreath for IMT. RHB program as CG.	Traditional exercise to enhance trunk control ability and included a stretching exercise for trunk flexibility, for 6 weeks, 5 days per week, once for 40 min.	-6-MWT -BBS
KM Jung, 2017 [25]	Korea	*n* = 12 GI: 6 GC: 6 41.6% men 61.7 years	**Inclusion:** Hemiparesis due to stroke, event occurring <6 months previously, ability to use a cycle ergometer, no restriction in lung function and no neurologic, orthopedic, or unstable cardiac condition, and ability to walk 100 m. **Exclusion:** Comorbidities or disabilities that would preclude study, and any uncontrolled health condition for which exercise is contraindicated.	*Session*: IMT 6 sets of 5 min each for 30 min a day, 5 times a week, for 4 weeks. Load: IMT at 30% MIP. *Device*: Threshold IMT. Moreover, traditional physical therapy and occupational therapy.	A self-selective intensity exercise with an ergonomic cycle for 30 min a day, five times a week, for four weeks.	-FVC -FEV1 -6-MWT
NJ Jung, 2017 [26]	Daegu, Korea	*n* = 20 GI: 10 GC: 10 NA	**Inclusion:** Stroke 6 months or longer; had no visual field defect and auditory sense; Scored at least 24 on the K-MMSE; independent sitting and gait; had no pulmonary embolus; no orthopedic problem, or unstable cardiac condition; had not undergone chest or abdominal surgery. **Exclusion:** NA	*Session*: 2 sets of 10 times IMT, taking a break of 10 s between each set. 5 times a week, for 6 weeks. Load of IMT: 80% of MIP *Device*: Respifit-S. Neuro Developmental Treatment as CG.	Neuro developmental treatment physical therapy for 30 min per time, 3 times a week, for 6 weeks.	-6-MWT
Guillen-Sola et al., 2017 [28]	Spain	*n* = 50 GI1: 16 GI2: 16 GC: 18 51.3% men 69.0 years	**Inclusion:** Subacute ischemic stroke within 1 to 3 weeks of inclusion and dysphagia. **Exclusion:** History of previous neurological diseases and/or cognitive impairment	*Session*: IMT/EMT, 5 sets of 10 breaths followed by 1 min of unloaded recovery breathing off the device. Twice a day, 5 days per week for 3 weeks, with the assistance of a therapist. Load: 30% of MIP/MEP and increased weekly at intervals of 10 cm H_2_O. *Device*: Orygen dual valve. RHB program as CG.	RHB program: Physical, occupational and speech therapy targeting specific impairments in mobility, activities of daily living, swallowing and communication skills.	-MIP -MEP
Oh et al., 2016 [27]	Korea	*n* = 23 GI: 11 GC: 12 56.5% men 70.6 years	**Inclusion:** Unilateral stroke occurred 6 months prior to the study the ability to perform breathing training for 30 min or longer; no sight impairment; modified Ashworth scale (MAS) score for upper and low extremities < 2; and MMSE-K score > 24. **Exclusion:** NA	*Session*: 15 times per set, 10 sets. Warm-up and cool-down exercises twice in each set, with a rest time of 60 s. 20 min per day, 3 times per week for 6 weeks. Load: IMT resistance 30% MIP. *Device: n*/A. RHB program as CG.	RHB program: 20-min session, 3 times/week, for 6 weeks. General breathing exercises, abdominal strengthening exercises, and general physical therapy.	-FVC -FEV1 -PEF -BBS
Chen et al., 2016 [29]	Taiwan	*n* = 21 GI: 11 GC: 10 38.1% men 65.5 years	**Inclusion:** Adults 20–85 years with congestive heart failure and stroke. **Exclusion:** MIP ≥70% predicted MIP, could not tightly place the lips over a mouthpiece, recent acute exacerbation of COPD, pneumothorax or large bullae on chest radiography, long-term use of oxygen therapy, recent lung surgery marked osteoporosis, unstable angina, decompensated CHF, or arrhythmia.	*Session*: IMT with a load of 30% of MIP (adjusted by 2 cm H_2_O/week), for 30 min, 5 times/week, for 10 weeks. *Device*: Respironics. RHB program as CG.	Conventional stroke RHB program, 5 days/week for 10 weeks.	-FVC -FEV1 -Dyspnea -MBI
Messaggi-Sartor et al., 2015 [30]	Spain	*n* = 77 GI: 39 GC: 38 57.8% men 66.5 years	**Inclusion:** Adults, first-ever ischemic stroke, time since stroke < 3 weeks, hemiparesis in upper or lower limb, and gave informed consent and followed study procedures. **Exclusion:** Cardiopulmonary disease, neurologic condition other than stroke, significant alcohol abuse, medical treatment with potential effect on muscle structure and function.	IMT/EMT with workload of 30% MIP/MEP. *Session*: 5 sets of 10 repetitions followed by 1–2 min of unloaded recovery, twice a day, 5 days per week, for 3 weeks. 15–20 breaths/min *Device*: Orygen-Dual valve.	RHB program: physical, occupational, and speech therapy sessions (3 h per day, 5 days a week, for 3 weeks). Sham IEMT without load and progression.	-MIP -MEP
Kulnik et al., 2015 [31]	United Kingdom	*n* = 82 GI1: 27 GI2:26 GC:25 57.3% men 64.4 years	**Inclusion:** Adults, < 2 weeks of stroke, 5–25 score in NIHSS, ability to give informed consent and follow study procedures. **Exclusion:** Blood pressure > 180/100 mmHg, angina, myocardial infarction, or acute heart failure within 3 months, pulmonary disease; and neurological conditions other than stroke.	EMT (GI1) and IMT (GI2) *Session*: 5 sets of 10 breaths each, with 1-min rests between sets. Load was 50% of MIP or MEP. These pressures were reassessed, and resistance readjusted weekly. *Device*: Respironics	Sham training: without load and progression.	-MIP -MEP
CY Kim et al., 2015 [32]	Germany	*n* = 37 GI1: 15 GI2: 13 GC: 12 45.94% men 59.1 years	**Inclusion:** Ischemic or hemorrhagic post-stroke hemiparesis; K-MMSE ≥ 26. **Exclusion:** Previous musculoskeletal abnormalities, confusion, neurological disorders, significant perceptual, cognitive, or communication impairments, COPD, and asthma.	*Session*: IMT/EMT 5 sets (10 repetitions/set), 1-min of rest after each set. 30 supervised sessions (5 × 15 min/week, 6 weeks). *Device*: Tri-ball Incentive Spirometer. RHB program as CG.	RHB program: stretching exercises of the limbs, therapist-guided techniques for facilitating the normal movement pattern. 1 h, 5 times a week.	-FEV1 -FVC
J Kim et al., 2014 [33]	Korea	*n* = 20 GI: 10 GC: 10 54 years	**Inclusion:** Hemiparesis due to stroke in previous 6 months, capable of comprehending commands and walking for at least 6 min with/without assistive device, had no previous cardiovascular or respiratory problems, no medications that would influence the metabolic or cardiorespiratory responses to exercise, no previous exercise training ventilator muscles and no bone deformities of the chest or spine. **Exclusion:** NA	*Session*: IMT/EMT during 20 min. Individually loaded and set to the breathing capacity of each patient. *Device*: Respifit-S.	Conventional exercise treatments for 30 min (including joint mobility, eccentric contraction, muscle strengthening, and walking exercise) followed by a 10-min rest. Full body workout machine for 20 min. 3 times/week, for 4 weeks.	-FEV1 -PEF -6-MWT -Dyspnea
Jung and Kim, 2013 [34]	Korea	*n* = 29 GI: 15 GC: 14 58.6% men 59 years	**Inclusion:** Hemiplegia secondary to stroke > 6 months and were undergoing general physical therapy. **Exclusion:** Innate thorax deformity, rib fracture, or had a disease related to lungs, kidneys, the endocrine system, orthopedics, or rheumatology.	*Session*: IMT with a load of 30% of MIP (adjusted gradually). 20 min, 3 times/week, for 6 weeks. *Device*: Respironics	Nothing	-FVC -FEV1 -PEF
Britto et al., 2011 [35]	Brazil	*n* = 21 GI: 9 GC: 9 52% men 54 years	**Inclusion:** Adults > 20 years; hemiparesis due to stroke, MIP < 90%; no facial palsy, able to use a cycle ergometer, had no restrictions in lung function and no neurologic, orthopedic, or unstable cardiac conditions, nonsmokers, showed no receptive aphasia, and had not undergone thoracic or abdominal surgery. **Exclusion:** Patients unable to perform the tests and used medications that could interfere with neuromuscular control or cause drowsiness.	*Session*: IMT 30 min, 5 times/week, for 8 weeks. Each session was divided into 6 series of 5 min each, with a 1-min rest interval between series. IMT with a load of 30% of MIP (adjusted every 2 weeks, according the new MIP value). *Device*: Threshold IMT.	Sham respiratory training.	-MIP
K Kim et al., 2011 [36]	Korea, Daegu	*n* = 27 GI: 13 GC: 14 37% men 57 years	**Inclusion:** Stroke occurred greater than 6 months ago. **Exclusion:** Pulmonary disorders, severe aphasia, and impairment of cognitive function.	*Session*: IMT/EMT, 30 min, 3 times/week, for 4 weeks. Load: 50–60% of VC and low frequency (12–13 breaths/ minute). *Device*: SpiroTiger. RHB program as CG.	Conventional stroke physical therapy program (30 min, 3 times/week, for 4 weeks).	-FVC -FEV1 -PEF
Sutbeyaz et al., 2010 [37]	Turkey	*n* = 45 GI1: 15 GI2: 15 GC: 15 53.3% men 61.8 years	**Inclusion:** Unilateral stroke with hemiparesis in previous 12 months, enough unilateral upper torso and extremity nerve function and strength to accomplish arm crank ergometry, ability to understand and follow simple verbal instructions, no previous history of cardiovascular or respiratory problems, no medication that would influence metabolic or cardiorespiratory responses to exercise. **Exclusion:** Chronic pulmonary and/or cardiac disease, clinical signs of cardiac and/or respiratory disease, impaired level of consciousness and evidence of gross cognitive impairment.	*Session*: 2 sets/day of 15 min each, six times a week, for 6 weeks. Load of IMT starting at a load of 40% MIP. It was gradually increased, 5–10% each session, to 60% of MIP as tolerated. *Device*: Threshold IMT. RHB program as CG.	Conventional stroke rehabilitation program, 5 days a week for 6 weeks.	-FVC -FEV1 -PEF -MIP -MEP -Dyspnea

IME, inspiratory muscle endurance; MIP, maximal inspiratory pressure; MEP, maximal expiratory pressure; CG, control group; IG, intervention group; NIHSS, National Institutes of Health Stroke Scale; RHB, rehabilitation; MBI, Barthel Index; BBS, Berg Balance Scale; K-MMSE, Korean Mini-Mental State Examination, IMT, inspiratory muscle training; EMT, expiratory muscle training; 6-MWT, 6 min walking test; NA, not available; RMT; respiratory muscle training; FVC, forced vital capacity; PEF: peak expiratory flow; TIS, trunk impaired scale; VC, vital capacity; MRS, Modified Rankin Scale; FEV1, first second forced expiratory volume.

**Table 2 ijerph-17-05356-t002:** Risk of bias and study quality on the PEDro Scale.

Study	Random Allocation	Concealed Allocation	Baseline Similarity	Subject Blinding	Therapist Blinding	Assessor Blinding	<15% Dropouts	Intention to-Treat Analysis	Between-Group Difference Reported	Point Estimate, Variability Reported	Total
Liaw et al., 2020 [19]	Y	Y	Y	N	Y	N	N	Y	Y	Y	7
Lee et al., 2019 [20]	Y	Y	Y	N	N	Y	N	N	Y	Y	6
Menezes et al., 2019 [21]	Y	Y	Y	Y	N	N	Y	Y	Y	Y	8
Cho et al., 2018 [22]	Y	Y	Y	N	N	Y	N	N	Y	Y	6
Yoo et al., 2018 [23]	Y	N	Y	N	N	N	Y	N	Y	Y	5
Lee et al., 2018 [24]	Y	N	Y	N	N	N	N	N	Y	Y	4
KM Jung et al., 2017 [25]	Y	Y	Y	N	N	N	Y	N	Y	Y	6
NJ Jung et al., 2017 [26]	Y	N	N	N	N	N	N	N	Y	Y	3
Guillen-Sola et al., 2017 [28]	Y	N	Y	N	N	Y	N	Y	Y	Y	6
Oh et al., 2016 [27]	Y	N	Y	N	N	N	Y	N	Y	Y	5
Chen et al., 2016 [29]	Y	N	Y	N	N	Y	N	N	Y	Y	5
Messagi-Sartor et al., 2015 [30]	Y	Y	Y	Y	N	Y	N	Y	Y	Y	8
Kulnik et al., 2015 [31]	Y	Y	Y	N	N	Y	N	Y	Y	Y	7
CY Kim et al., 2015 [32]	Y	N	Y	N	N	Y	N	N	Y	Y	5
J Kim et al., 2014 [33]	Y	N	Y	N	N	N	N	N	Y	Y	4
Jung and Kim 2013 [34]	Y	N	Y	N	N	N	N	N	Y	Y	4
Britto et al., 2011 [35]	Y	Y	Y	N	N	Y	Y	N	Y	Y	7
K Kim et al., 2011 [36]	Y	N	Y	Y	N	Y	N	N	Y	Y	6
Sutbeyaz et al., 2010 [37]	Y	Y	Y	N	N	Y	Y	N	Y	Y	7

PEDro: Physiotherapy Evidence Database (www.pedro.org). Y: Yes; N: No.

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
