# Peer review of "Effectiveness of Respiratory Muscle Training for Pulmonary Function and Walking Ability in Patients with Stroke: A Systematic Review with Meta-Analysis"

_ijerph, 2020, doi:10.3390/ijerph17155356_

Round 1

Reviewer 1 Report

One of the post-stroke dysfunctions is the respiratory muscles weakness causing breathing problems. From this point of view, any respiratory muscle interventions are of high importance.

The presented manuscript provides the results of systematic review and meta-analysis aimed to synthetize the most novel evidence about the effectiveness of RMT to improve respiratory function, respiratory muscle strength and functional capacity in post-stroke patients.

The study is well performed and written. However, I a few minor comments to the Authors.

  1. Although the Authors cite the appropriate method for statistical heterogeneity analysis, I’m interested, how they deal with results in-between the groups, e.g. 85%? Did they considered it as substantial or considerable? (lines 121-124).
  2. The Table footer should be below the Table instead of inside the Table.
  3. There are few grammar errors that should be corrected (e.g. “decreases in respiratory volumes” instead of “decreases of respiratory volumes”, lines 42-43; “the mean age” instead of “the age mean”, line 156 etc.).

Author Response

We would like to thank you for giving us the opportunity to revise and improve our manuscript; we also thank the reviewers for the thoughtful and constructive comments. We have considered all the suggestions and have incorporated them into the revised manuscript, and as a result, we believe our manuscript is stronger. An itemized point-by-point response to the reviewers’ comments has been uploaded.

REVIEWER 1

One of the post-stroke dysfunctions is the respiratory muscles weakness causing breathing problems. From this point of view, any respiratory muscle interventions are of high importance.

The presented manuscript provides the results of systematic review and meta-analysis aimed to synthetize the most novel evidence about the effectiveness of RMT to improve respiratory function, respiratory muscle strength and functional capacity in post-stroke patients.

The study is well performed and written. However, I a few minor comments to the Authors.

  1. Comment: Although the Authors cite the appropriate method for statistical heterogeneity analysis, I’m interested, how they deal with results in-between the groups, e.g. 85%? Did they consider it as substantial or considerable? (lines 121-124).

Authors: Thank you for this interesting comment. As suggested in the Cochrane Collaboration Handbook, the use of thresholds for the interpretation of I2 can be misleading because the importance of inconsistency depends on several factors such as the magnitude and direction of effects and p value, which must be taken into account. Thus, the authors consider that it is not so important to classify heterogeneity in one category, but to investigate and address the effects of heterogeneity with different strategies such as: random-effects models, appropriate effect measures, sensitivity analysis…

  1. Comment: The Table footer should be below the Table instead of inside the Table.

Authors: Thank you. We have correctly inserted the table footer (Table 1).

  1. Comment: There are few grammar errors that should be corrected (e.g. “decreases in respiratory volumes” instead of “decreases of respiratory volumes”, lines 42-43; “the mean age” instead of “the age mean”, line 156 etc.).

Authors: Checked and modified. Thank you.

Line 42-43: “decreases in respiratory volumes and flows…”

Line 106: “The total PEDro score ranges from 0 to 10 points (the first item…”

Line 156: “The mean age of participants was 61.3 years.”

Reviewer 2 Report

Summary:

A meta-analysis of the literature concerning respiratory muscle training was conducted to examine the effectiveness of these techniques in randomized controlled trials. From the 800+ articles that were reviewed, 19 met the inclusion criteria.

Comments:

The authors have presented a meta-analysis to review the effects of techniques used to improve respiratory function in post-stroke patients. Overall, the assessment performed by the authors has been performed with great detail, however this meta-analysis did not consider the severity of stroke in their analysis, which may have biased their study – especially when improvements in respiratory variables is concerned.

Introduction:

The authors have not defined what respiratory muscles were that they looked at: diaphragm, external/internal intercostals, scalenes, superior/inferior serratus posterior. Why is this important? Because the nerval signaling to the diaphragm is independent of the neural network invoking the stimulation of the other (supplementary) muscles groups and may be affected differently depending upon the stoke/stoke severity.

In lines 59-61, the authors indicate they are performing a systematic review of novel evidence in the literature since the last meta-analysis of this topic, dated 2018. However, in the Methods section (lines 69-78), the authors indicate that searchers were conducted from the inception to the most recent publications. Why not perform searches in a defined window from the last 5-6 years if the authors were looking for novel evidence?

Lines 46-52: these are independent sentences, not paragraphs, and would be better served combined into one paragraph

Lines 55-56: this sentence is incomplete – what are the authors suggesting?

Lines 56-58: provide references for this statement

Results:

Lines 1-4: this statement needs clarity. Should the text read “four out of twenty studies”?

Table 2.: the column headings need to be formatted so that the variable is clear to the reader

Author Response

We would like to thank you for giving us the opportunity to revise and improve our manuscript; we also thank the reviewers for the thoughtful and constructive comments. We have considered all the suggestions and have incorporated them into the revised manuscript, and as a result, we believe our manuscript is stronger. An itemized point-by-point response to the reviewers’ comments has been uploaded.

REVIEWER 2

Summary:

A meta-analysis of the literature concerning respiratory muscle training was conducted to examine the effectiveness of these techniques in randomized controlled trials. From the 800+ articles that were reviewed, 19 met the inclusion criteria.

  1. Comment: The authors have presented a meta-analysis to review the effects of techniques used to improve respiratory function in post-stroke patients. Overall, the assessment performed by the authors has been performed with great detail, however this meta-analysis did not consider the severity of stroke in their analysis, which may have biased their study – especially when improvements in respiratory variables is concerned.

Authors: Thanks for your interesting comment. It would have been very interesting to perform a subgroup analysis by stroke severity categories, however it was not possible due to the lack of this information in the original studies.

  1. Comment: The authors have not defined what respiratory muscles were that they looked at: diaphragm, external/internal intercostals, scalenes, superior/inferior serratus posterior. Why is this important? Because the nerval signaling to the diaphragm is independent of the neural network invoking the stimulation of the other (supplementary) muscles groups and may be affected differently depending upon the stoke/stoke severity.

Authors: Thank you, this appreciation is very interesting. However, the aim of this paper, as well as the original studies included in it, was to examine the effectiveness of respiratory muscle training in general, not the specific work of a single muscle or in a specific degree of stroke severity. Therefore, although the suggestion is interesting, we consider that it is not possible to specify the trained muscles since this variable was not reported in the studies included in this meta-analysis.

  1. Comment: In lines 59-61, the authors indicate they are performing a systematic review of novel evidence in the literature since the last meta-analysis of this topic, dated 2018. However, in the Methods section (lines 69-78), the authors indicate that searchers were conducted from the inception to the most recent publications. Why not perform searches in a defined window from the last 5-6 years if the authors were looking for novel evidence?

Authors: Thank you. Although the strategy suggested by the reviewer could be quicker and easier, we believe that a systematic search of all available evidence is more appropriate, since previous meta-analyses may not have had the same eligibility criteria as we have established, which could result in a loss of studies.

  1. Comment: Lines 46-52: these are independent sentences, not paragraphs, and would be better served combined into one paragraph.

Authors: Thank you. We have combined these sentences as follows:

Lines 46- 52: “Respiratory muscles respond to training similarly to any other skeletal muscle, so just as the upper and lower limb muscles are trained in stroke patients, the respiratory muscles must be trained. In this regard, respiratory muscle training (RMT) consists of repetitive breathing exercises…”

  1. Comment: Lines 55-56: this sentence is incomplete – what are the authors suggesting?

Authors: Thank you. We have modified the sentence for clarity as follows:

Lines 55- 57: “However, some weaknesses of these reviews were that they not only included randomised controlled trials (RCTs) [10], and that they included a limited number of studies [3, 8]. Moreover, since the last systematic review and meta-analysis [7]…”

  1. Comment: Lines 56-58: provide references for this statement.

Authors: Thank you.

Lines 55- 57: “However, some weaknesses of these reviews were that they not only included randomised controlled trials (RCTs) [10], and that they included a limited number of studies [3, 8].

  1. Comment: Lines 1-4: this statement needs clarity. Should the text read “four out of twenty studies”?

Authors: Thank you. We have modified this sentence for clarity as follows:

Lines 1-3: “In the control group, most studies conducted the conventional stroke rehabilitation program, however four out of twenty studies performed sham respiratory training without resistance and/or progression [21, 30, 31, 35].”

  1. Comment: Table 2. The column headings need to be formatted so that the variable is clear to the reader.

Authors: Thank you very much. As suggested, we have edited the format of the column headings (Table 2).

We sincerely appreciate the fruitful reviewers’ comments.

Round 2

Reviewer 2 Report

The authors have sufficiently addressed the concerns of this author